# Development of Forensically Important *Sarcophaga peregrina* (Diptera: Sarcophagidae) and Intra-Puparial Age Estimation Utilizing Multiple Methods at Constant and Fluctuating Temperatures

**DOI:** 10.3390/ani13101607

**Published:** 2023-05-11

**Authors:** Yanjie Shang, Fengqin Yang, Fernand Jocelin Ngando, Xiangyan Zhang, Yakai Feng, Lipin Ren, Yadong Guo

**Affiliations:** 1Department of Forensic Science, School of Basic Medical Sciences, Central South University, Changsha 410013, China; 2Department of Forensic Medicine, School of Basic Medical Sciences, Xinjiang Medical University, Urumqi 830017, China

**Keywords:** *Sarcophaga peregrina*, intra-puparial age estimation, constant and fluctuating temperatures, DGEs, ATR-FTIR, CHCs

## Abstract

**Simple Summary:**

Constant and fluctuating temperatures can affect the growth and development of insects, which will affect the accuracy of minimum postmortem interval estimation. Our study found that the development time of *S. peregrina* was prolonged under fluctuating temperatures. The estimation of the pupal stage has a significant effect on the estimation of PMI. We explored the feasibility of different methods for intra-puparial age estimation and found that the DEG expression profile, ATR-FTIR, and CHCs detection, combined with chemometrics, had the potential to estimate the intra-puparial age of *S. peregrina* under constant and fluctuating temperatures.

**Abstract:**

*Sarcophaga peregrina* (Robineau-Desvoidy, 1830) has the potential to estimate the minimum postmortem interval (PMI_min_). Development data and intra-puparial age estimation are significant for PMI_min_ estimation. Previous research has focused on constant temperatures, although fluctuating temperatures are a more real scenario at a crime scene. The current study examined the growth patterns of *S. peregrina* under constant (25.75 °C) and fluctuating temperatures (18–36 °C; 22–30 °C). Furthermore, differentially expressed genes, attenuated total reflectance Fourier-transform infrared spectroscopy, and cuticular hydrocarbons of *S. peregrina* during the intra-puparial period were used to estimate age. The results indicated that *S. peregrina* at fluctuating temperatures took longer to develop and had a lower pupariation rate, eclosion rate, and pupal weight than the group at constant temperatures did. Moreover, we found that six DEG expression profiles and ATR-FTIR technology, CHCs detection methods, and chemometrics can potentially estimate the intra-puparial age of *S. peregrina* at both constant and fluctuating temperatures. The findings of the study support the use of *S. peregrina* for PMI_min_ estimation and encourage the use of entomological evidence in forensic practice.

## 1. Introduction

Entomology evidence could be the only evidence that can be used to estimate the minimum postmortem interval (PMI_min_) in decaying corpses. After insects are collected from the corpse and identified, the PMI_min_ is estimated using development data of necrophagous flies [1,2]. Therefore, development data collection and accurate age estimation of necrophagous flies are meaningful for PMI_min_ estimation.

Insects are poikilothermic animals. The temperature has a significant impact on the development and growth of insects [3,4]. Previous studies that used necrophagous insects to estimate PMI_min_ focused on constant temperatures. These studies focused on several necrophagous Dipterans (e.g., *Chrysomya megacephala* (Fabricius, 1794) [5], *Sarcophaga dux* (Thomson, 1869) [6], *Lucilia cuprina* (Wiedemann, 1830) [7], *Megaselia scalaris* (Loew, 1866) [8], and *Musca domestica* (Linnaeus, 1758) [9]), Coleoptera (e.g., *Necrobia rufipes* (De Geer, 1775) [10]), and Hymenoptera (e.g., *Nasonia vitripennis* (Walker, 1836) [11]). However, fluctuating temperatures are more realistic than constant temperatures are [1,12,13,14]. Exploring the development and growth of necrophagous flies at fluctuating temperatures is more important for PMI_min_ estimation. Wu et al. [15] concluded that the impact of constant and fluctuating temperatures on the development and growth of flies varied from study to study, with some studies accelerating development and others retarding development. For example, temperature fluctuations decelerated the development of *Scathophaga stercoraria* (Linnaeus, 1785) [16], *Aldrichina grahami* (Aldrich, 1930) [12], *Calliphora vicina* (Robineau-Desvoidy, 1830), and *Calliphora vomitoria* (Linnaeus, 1758) [13]. However, the authors of studies about *Sarcophaga argyrostoma* (Robineau-Desvoidy, 1830), *Lucilia illustris* (Meigen, 1826) [13] and *Protophormia terraenovae* (Robineau-Desvoidy, 1830) [14] found that the development was accelerated. The explanation for this inconsistent result is that when the rearing temperature falls within the moderate temperature range of the insect, the development time of the insect appears to be the same at fluctuating and constant temperatures. The development times differed when rearing temperatures were below or near the lower developmental threshold temperature or above or near the higher fatal developmental threshold temperature [17,18]. For more accurate data to determine PMI_min_, it is required to gather development data from necrophagous flies under fluctuating temperatures.

The intra-puparial period, which occupies half of the immature stage [19], is crucial for estimating PMI_min_. Many methods for estimating pupal age have been reported, including puparium color variation [20,21], intra-puparial morphological characteristics [22], x-irradiation [23], micro-computed tomography imaging [24], and optical coherence tomography [25]. However, morphological methods require morphological knowledge, and the application of development data in the intra-puparial period is limited. Micro-CT techniques, for example, are uncommon at forensic workstations and inconvenient to use at crime scenes. Therefore, it is crucial to find a fast, accurate, and convenient method for estimating the intra-puparial period age.

Studying differentially expressed genes (DEGs) as a method for age estimation has been used for blow flies and flesh flies [26,27]. Fourier-transform infrared (FTIR) spectroscopy can determine the metabolic properties of insects based on absorption characteristics, such as lipids, proteins, and cellular processes [28]. Attenuated total reflection Fourier-transform infrared (ATR-FTIR) has been used to evaluate the effects of the cause of death on PMI estimation [29], to discriminate between normal and pathological populations [30,31,32] and to detect SARS-CoV2 infections on the spot [33]. However, there are a limited number of reports of FTIR applications in forensic entomology. Cuticular hydrocarbons (CHCs), the major components of the waxy layer of the insect cuticle [34,35], which has been proved to be a reliable method of predicting the age of the fly [36,37,38,39]. As a result, further research into the value of DEGs, ATR-FTIR, and CHCs in intra-puparial age estimation is required.

*Sarcophaga peregrina* (Robineau-Desvoidy, 1830) (Diptera: Sarcophagidae) is a valuable flesh fly species of medical, veterinary, ecological, and forensic entomological importance [40,41,42]. It plays a crucial role in decomposing cadavers of forensic significance, especially in PMI_min_ estimation. Studies associated with *S. peregrina* are limited in number [43], and they largely focus on larval morphology [44,45], molecular research [46,47], cuticular hydrocarbon profile investigation [48,49,50], development data collecting [19], chromosome-level research [51,52], and effects of drugs and heavy metals [53,54,55]. Until this study, we have not searched the literature on the effect of fluctuating temperature on the development of *S. peregrina* through commonly used literature search platforms (PubMed and Web of Science).

In the current study, developmental data of *S. peregrina* were collected at constant and fluctuating temperatures for PMI_min_ estimation. In addition, six DEG expression profiles, the ATR-FTIR spectroscopy characters, and the CHC profiles of *S. peregrina* were investigated and analyzed during the intra-puparial period. This study provides a basis for rapid and accurate PMI_min_ estimation.

## 2. Materials and Methods

### 2.1. Insect Rearing and Experimental Temperature Settings

*Sarcophaga peregrina* was captured using pig carcasses in Changsha (28°12′ N, 112°58′ E), Hunan Province, China, in July 2017 and reared in Guo’s Lab (Changsha China). Traditional morphological characteristics [56] the long cytochrome oxidase subunit I (CO I) sequences were used to identify the species. Adults were fed in an artificial climate chamber (LRH-250-GSI, Taihong Co., Ltd., Shaoguan, China) with 25.0 ± 1.0 °C, 75% RH, a 12 h light and 12 h dark cycle, and the feeding conditions were fresh water and mixed milk powder and sugar.

The experimental constant and fluctuating temperatures (Figure 1) were designed based on the temperature variations in Changsha, China, from June to October 2011 to 2021 (Appendix A). The temperature groups were recorded as fluctuating model A (group A: 18–36 °C), fluctuating model B (group B: 22–30 °C), and constant model C (group C: 25.75 °C). The temperature tolerance of the artificial climate chamber was 1 °C with 75% RH and a 12 h: 12 h (L: D) cycle, as shown in Figure 1 and Appendix A, respectively.

### 2.2. Collection of Samples for Developmental Analysis and Multi-Method Analysis

A pig lung was used to induce the larvae to hatch. Larvae (ca. 1500) produced within 1 h were divided into three groups: A, B, and C. Ten larval samples were randomly taken every eight hours. Larval instar was identified using a Zeiss 2000-C stereomicroscope (Carl Zeiss, Jena, Germany). Larval length was measured using a digital caliper (Mineette, Shanghai, China). The developmental time, pupation rate, eclosion rate, and pupal weight were recorded during the experiment. Data were analyzed using Origin Pro 8.6 software (OriginLab, Northampton, MA, USA, SCR: 015636). Three replications were performed.

Larvae produced within 1 h were cultured at constant temperature (group C) and fluctuating temperature (group A). Thirty pupae were harvested at 24 h intervals until the emergence of adults. The samples collected each time were divided into three groups on average for DEGs, ATR-FTIR, and CHCs analysis, respectively. Three replications were performed. A total of 1620 pupae were collected: 810 in group A and 810 in group C.

### 2.3. DEGs Study

Total RNA from each intra-puparial tissue obtained by removing the puparium was extracted using the SteadyPure Quick RNA Extraction Kit (Code No. AG21017) (Accurate Biotechnology (Hunan) Co., Ltd. Changsha, China) and stored at −80 °C following the manufacturer’s instructions. Total RNA quantification was performed using NanoDrop 2000 (Thermo Fisher Scientific, Wilmington, DE, USA). The ratios of optical density (OD) of 260/280 and 260/230 were selected as the evaluation indicators. cDNA was synthesized with the Evo M-MLV RT Mix kit (Code No. AG11728) (Accurate Biotechnology (Hunan) Co., Ltd. Changsha, China), and then qPCR was performed.

The candidate DEGs were identified according to the genome [51] (NCBI no. JABZEU000000000) and transcriptome data [57] (NCBI nos. PRJNA795032 and PRJNA810308) of *S. peregrina*. Quantitative real-time PCR operation was performed according to the instructions of the SYBR^®^ Green Premix Pro Taq HS qPCR Kit (ROX Plus) (Code No. AG11718) (Accurate Biotechnology (Changsha, China) Co., Ltd.) on an ABI 7500 Real-Time PCR system (Applied Biosystem, Carlsbad, CA, USA). Primers for the DEGs and reference genes were designed using Primer Premier 5.0 (Biosoft Premier, Palo Alto, CA, USA) and are listed in Appendix A. The 2^−△△Ct^ method was used to calculate the relative quantification (RQ) of the DEGs. All statistical analysis was conducted using the software Origin Pro 8.6.

### 2.4. ATR-FTIR Study

The intra-puparial tissue was ground using liquid nitrogen, and later, transferred into a 1.5 mL EP tube and shaken using a vortex mixer (Guangzhou GenXion Biotechnology Co. LTD, Guangzhou, China). An 850 Fourier-transform infrared spectrometer equipped with an ATR attachment was used (Tianjin Gangdong Technology Development Co. LTD, Tianjin, China) to collect spectra. The ATR attachment was cleaned with absolute ethanol before each measurement, and background spectra were acquired before each operation. Spectra were recorded at a resolution of 4 cm^−1^ in the frequency range from 4000 to 900 cm^−1^, with 32 scans per sample. OMNIC, version 9.2 (Thermo Fisher Scientific, Wilmington, MA, USA), was used to process the spectral data. Unnecessary background information was reduced by spectral preprocessing. The standard normal variate (SNV) was used to decrease the impact of light scattering and spectral size due to sample thickness [58]. Furthermore, all spectral data were smoothed and denoised (Savitzky–Golay convolution algorithm; number of smoothing points: 15). Finally, the spectral range of 1800–900 cm^−1^ was used for further analysis.

In this study, Principal Component Analysis (PCA), Orthogonal Partial Least Squares Discriminant Analysis (OPLS-DA), and Partial Least Squares Discriminant Analysis (PLS) were performed using SIMCA 14.1 software to observe the spectral characters of the intra-puparial tissue at different times of sampling. PCA can reduce the dimensions of the data and retain original data features. OPLS-DA is a supervised statistical method of discriminant analysis [59,60]. For OPLS-DA, values of R^2^ (cum) and Q^2^ (cum) close to 1.0 indicate an excellent model in theory, >0.5 indicates a better model, and >0.4 indicates an acceptable model. The permutation test was used to examine if the model was overfitted. PLS regression analysis can decompose X variable and extract latent variable under the guidance of Y variable [61,62]. R^2^ and the root mean square error (RMSE) were used to evaluate the PLS model; values close to 1 for R2 and 0 for RMSE indicate a better model. The variable importance in projection (VIP) was used to assess the contribution of each variable to the OPLA-DA and PLS models, and a VIP value of >1.0 was considered to be significant.

### 2.5. CHCs Study

Referring to Zhang et al.’s [49,57] operation, the intra-puparial tissue was immersed in a 2 mL glass vial containing 1 mL of redistilled hexane for 30 min at room temperature. An alkane mixture of heptane and tetracontane (C7–C40, 1 μg/mL, O2SI) dissolved in 1 mL of hexane was selected as an external standard. Using a syringe with a nylon membrane and filter, the soaking solution was transferred into a new 2 mL glass vial. Following that, the samples were vacuum concentrated and dried before being dissolved in 200 μL redistilled hexane for GC-MS analysis. GC–MS (Agilent Technologies, 7890B–5977A GC/MSD) fitted with a DB-5MS capillary column was used to analyze the CHC. Operations of sample loading were performed and instrument condition settings were used according to references [49] that have already been reported. Due to the length limitation, specific setting contents are not described here.

MSD ChemStation Data Analysis F.01.03 was used to integrated the peak areas, and the qualitative identification of CHC was carried out by comparing the chromatographic retention time, characteristic fragment ions, and abundance ratio with those of standard values. The Kováts retention index, NIST 11 library, and literature [49,63,64,65,66,67,68,69,70] were selected to assist identification, which we called the CHCs database. PCA, OPLS-DA, and PLS models were used to visualize the filtered data.

## 3. Results

### 3.1. Development Analysis

The correlation between larval body length and development time under different temperatures is shown in Figure 2. The model of fourth-order polynomials was used to establish nonlinear regression. Table 1 shows the equations and R^2^ values of the model. The results show that the larval body length of group C was longer than those of both group A and group B before 80–90 h, and the longest larvae can be observed in group C.

*S. peregrina* could complete their life history under constant and fluctuating temperatures, and the development duration is listed in Table 2. Group C spent the minimum amount of time reaching full development. The development rate of group A was similar to that of group B and slower than that of group C. As shown in Figure 3, the pupariation rate, eclosion rate, and pupal weight were lower and statistically significant in groups A and B than they were in group C. These results indicate that fluctuating temperatures can affect the development of *S. peregrina*.

### 3.2. DEGs Analysis

The expression profiles of six DEGs (circRNA_2143, circRNA_3489, circRNA_2847, fln, UQCRFS1, and COX5A) changed with the increase in intra-puparial age (Figure 4). Polynomial regression analysis described the relationship between the level of DEG expression of *S. peregrina* and developmental age (days) (Table 3). The increasing expression tendencies of the six DEGs, which are similar under constant and fluctuating temperatures, indicate that these six DEGs can be gene marker candidates to estimate the intra-puparial age of *S. peregrina* at both constant and fluctuating temperatures.

### 3.3. ATR-FTIR Analysis

Figure 5 shows the average FTIR spectra of different intra-puparial development ages (days) in the range of 1800–900 cm^−1^. Based on previous studies [71,72,73], the main infrared absorption band identification is shown in Table 4. The OPLS-DA classification model (Figure 6) showed that the same day’s spectra were well clustered and distinguished from those of the other days for both groups, except for a few data overlaps between day 5 and day 9 for group C. Common wavenumber bands with VIP > 1 (Appendix A) were 989.31–898.672 cm^−1^ and 1716.35–1675.85 cm^−1^. The model evaluation parameters of the PLS are listed in Table 5: R^2^ > 0.9 and RMSE of 0.4271 day and 0.7683 day for groups A and C, respectively. Figure 6 shows the regression model to predict the intra-puparial age. Wavenumbers with VIP > 1 are shown in Appendix A. The common wavenumber bands were 1760.7–1754.92 cm^−1^, 1729.85–1681.64 cm^−1^, 1660.42–1616.07 cm^−1^, 1544.71–1506.14 cm^−1^, 1033.67–1031.74 cm^−1^, 1022.09–1016.31 cm^−1^, and 960.384–898.672 cm^−1^. These results suggest that the spectral differences observed in this study were caused by changes in proteins, lipids, and other intra-puparial tissue components. The error of intra-puparial age inference is smaller in a fluctuating temperature environment.

### 3.4. CHC Analysis

A total of 41 hydrocarbons were identified, including 21 n-alkanes, 17 branched alkanes, and 3 alkenes with a carbon chain length between C9 and C32 (Table 6). According to the mini bar chart, branched alkanes were the most abundant compound from day 1 to day 8 at constant or fluctuating temperatures; on day 9, it was n-alkanes. The CHCs profiles (Appendix A) show that the most abundant compounds in group A were 4-Methyl C17 on days 1–3 and DiMethyl C29 on days 6–9. Additionally, a transitional stage could be observed on days 4–5. Except for a slightly different transitional stage on days 4–5, group C showed nearly the same change as group A did. In addition, dominant CHCs in the intra-puparial tissues showed a shift from a low to a high molecular weight with the age of the pupae. The OPLS-DA plot (Figure 7) shows that the clustering effect of group A was better than that of group C. Common variables (Appendix A) of groups A and C with VIP > 1 (Appendix A) were selected. 4-Methyl C10, Methyl C29, C19, C21, and 7-Methyl C27 were identified. The parameters of PLS analysis are listed in Table 5: R^2^ > 0.9, and RMSEs of 0.4837 day and 0.7653 day for groups A and C, respectively. C27, C25, C23, and DiMethyl C29 were identified. The CHCs identified above suggest a possible link between changes in the most abundant CHCs and increasing intra-puparial age, particularly DiMethyl C29, which could be useful in estimating *S. peregrina* intra-puparial age at constant and fluctuating temperatures. Additionally, the error of intra-puparial age inference is smaller in a fluctuating temperature environment.

## 4. Discussion

Fluctuating temperatures can influence the development of flies. In studies, temperature fluctuations have been shown to either accelerate or slow the development of insects [13,14] or have no significant difference in development [74]. This study found that fluctuating temperatures influenced the feeding stage of *S. peregrina*. It led to smaller lengths and weights, longer developmental times, and a lower survival rate for larvae and pupae. The thermal mean and its proximity to the developmental threshold [17,18] could be the main reasons for this phenomenon. The developmental rate difference between fluctuating and constant temperatures grows as the temperature fluctuation range increases [4]. When the range is within the suitable range for insect growth, the effect of constant and fluctuating temperatures will not be significant. However, when the range exceeds the suitable range, such as a lower or higher temperature, the effects of fluctuating temperature are relatively large [75]. Changes in the development time at fluctuating temperatures can lead to incorrect age estimations, which in turn leads to incorrect PMI_min_ estimations.

More development data under fluctuating temperatures are required to accurately determine the intra-puparial age of necrophagous flies. However, this can greatly increase forensic entomologists’ workload. Therefore, it is crucial to develop an approach that is faster, simpler, more accurate, and universally applicable.

The value of DEGs for inferring the age of flies has been confirmed [27,57,76]. Studies on *Sarcophaga dux* [6], *Aldrichina graham* [77], *Lucilia illustris* [78], *Glossina morsitans* [79], and *Calliphora vicina* [1,80,81] showed that DEGs can be used to determine the age of necrophagous flies. However, the expression trends of the DEGs in the above research were not singular, and most of them were performed under constant temperatures. DEGs that can stably express at constant and fluctuating temperatures and show a single temporal expression trend are expected to be genetic markers for age estimation. For example, a study on the expression of bcd, sll, and cs shows a significant linear trend during the development of blowfly eggs [27]. Candidate DEGs for *S. peregrina* were chosen from previously published results in our previous study [26]. Their expression trends were not singular, and they were demonstrated under constant temperatures. The candidate DEGs in this study were chosen from *S. peregrina*’s reference genome and transcriptome data [49,51]. We found that the six DEGs showed a single up-regulated temporal expression, with the intra-puparial period increasing under constant and fluctuating temperatures. This result suggested that the six DEGs could become insect gene markers for estimating the intra-puparial age of *S. peregrina*. Notably, three circular RNAs (circRNAs), which are more stable than linear RNAs are [82,83], were introduced as candidate DEGs for the first time in this paper. Although DEGs can be used to estimate the PMI_min_, the number of studies on the profiles of necrophagous flies is still limited. More research should be conducted.

The amount of research on the application of vibrational spectroscopy in biochemistry and forensic science is increasing [32,84,85]. Based on vibrational bands, the biochemical information associated with the fingerprint region (1800–900 cm^−1^) depicts the structure and function of cell samples [86,87,88,89]. With the enormous quantity of data produced by IR spectroscopy, chemometrics can help with data pre-processing, sample classification, and reducing the dimensionality of chemical information [84]. Perez-Mendoza [90] and Pickering et al. [91] successfully identified the age of different species by using infrared spectroscopy. Our study found that the proteins, lipids, and other components of the intra-puparial tissues, represented by the common bands, were affected by the development of flies. These substances are important for the model’s establishment and age estimation. The model parameters suggested that better results were obtained at fluctuating temperatures, demonstrating that temperature settings that match the real environment are crucial for accurate age estimation. Moreover, the automation and convenience of the test instruments make immediate analysis at crime scenes possible [86]. The use of spectroscopic methods in forensic science could improve the use of entomological evidence.

Cuticular hydrocarbons are the main component of the waxy layer of insect cuticles [35] and play a key role in insect survival. They can act as a barrier for insects to prevent water loss, avoid mechanical damage, isolate microorganisms and chemicals, and they also act as pheromones for insects [35,92,93,94]. *Sarcophaga bullata* CHC profiles during development have been published [95]. Studies on *Chrysomya albiceps* [38], *Chrysomya rufifacies* [37], and *Sarcophaga peregrina* [49] also confirmed that the dynamics of CHC profiles during the development of flies can be used to determine the age of necrophagous flies [96]. The carbon chain length of the CHCs identified in this study was between C9 and C32, which was within the range of C8–C36 discovered by Zhang et al. [49]. However, the most abundant CHC differed from that in Zhang et al.’s study [49]. This is because the tissue used in the study, intra-puparial tissue, differs from the pupal shell used in their study. Meanwhile, as the intra-puparial age increased, the dominant CHCs shifted from low- to high-molecular-weight hydrocarbons. This is similar to the study of *Chrysomya rufifacies* [37]. The next stage of *S. peregrina* after completing metamorphic development is the adult stage. The heavier, long-chain hydrocarbons, which are the most important compounds in the insect cuticle [48,97], are essential for waterproofing [98,99,100] and help adults to survive in complex external environments. Additionally, the model parameters showed that better results could be obtained from group A, which was placed under fluctuating temperatures. These results indicate that more accurate data can be obtained for precise age estimation in a more realistic simulated situation. Additionally, CHC’s profiles can be used to determine the intra-puparial age of *S. peregrina*, as well as the PMI_min_.

## 5. Conclusions

This study investigated methods for estimating the intra-puparial age of *Sarcophaga peregrina* under constant and fluctuating temperatures, such as DEGs, ATR-FTIR, and CHC profiles. We found that the development of *S. peregrina* was different under constant and fluctuating temperatures and demonstrated that the three methods are meaningful for pupal age estimation.

## Figures and Tables

**Figure 1 animals-13-01607-f001:**
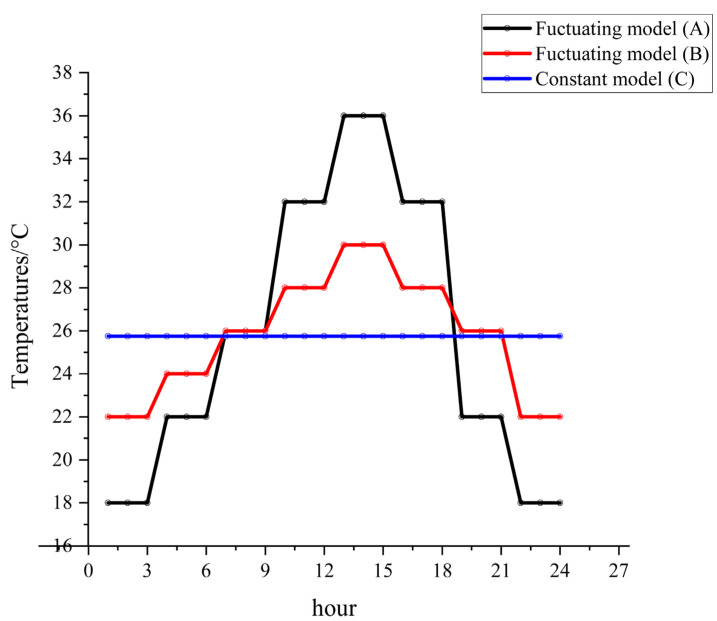
The specific temperature settings of the artificial climate chamber.

**Figure 2 animals-13-01607-f002:**
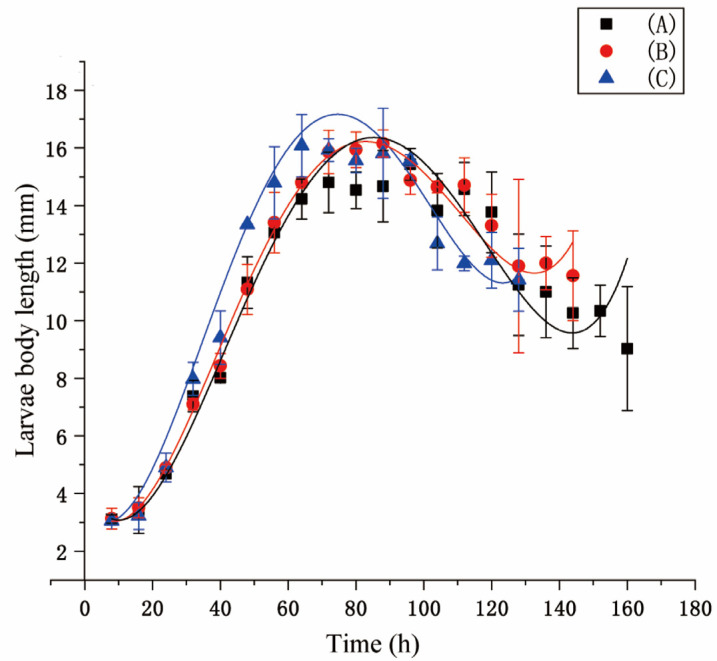
Nonlinear regression of larval body length changes under different temperature conditions. (A represents the fluctuating model A, B represents the fluctuating model B, and C represents the constant model C).

**Figure 3 animals-13-01607-f003:**
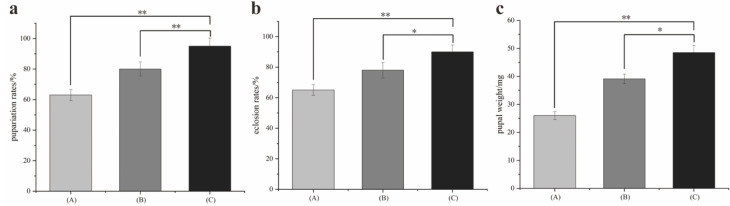
The pupariation rate, eclosion rate, and pupal weight were lower and statistically significant in groups A and B than they were in group C. (**a**) Pupariation rate. (**b**) Eclosion rate. (**c**) Pupal weight (* represents *p* < 0.05, ** represents *p* < 0.01).

**Figure 4 animals-13-01607-f004:**
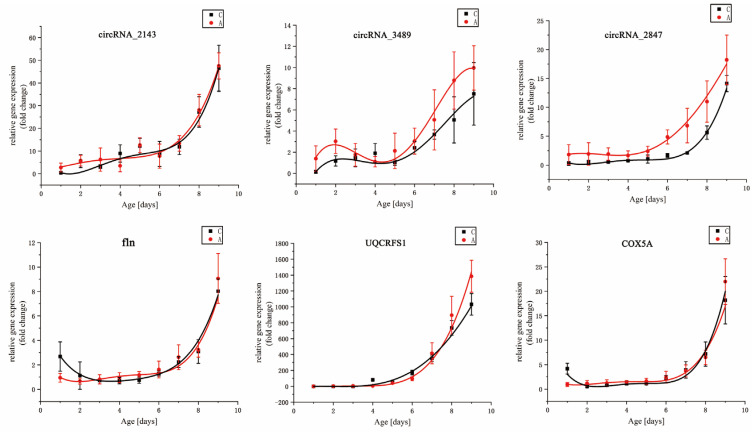
The expression levels of the six DEGs at constant (group C at 25.75 °C) and fluctuating temperatures (group A at 18–36 °C) in correlation to the intra-puparial age (days) of *S. peregrin*.

**Figure 5 animals-13-01607-f005:**
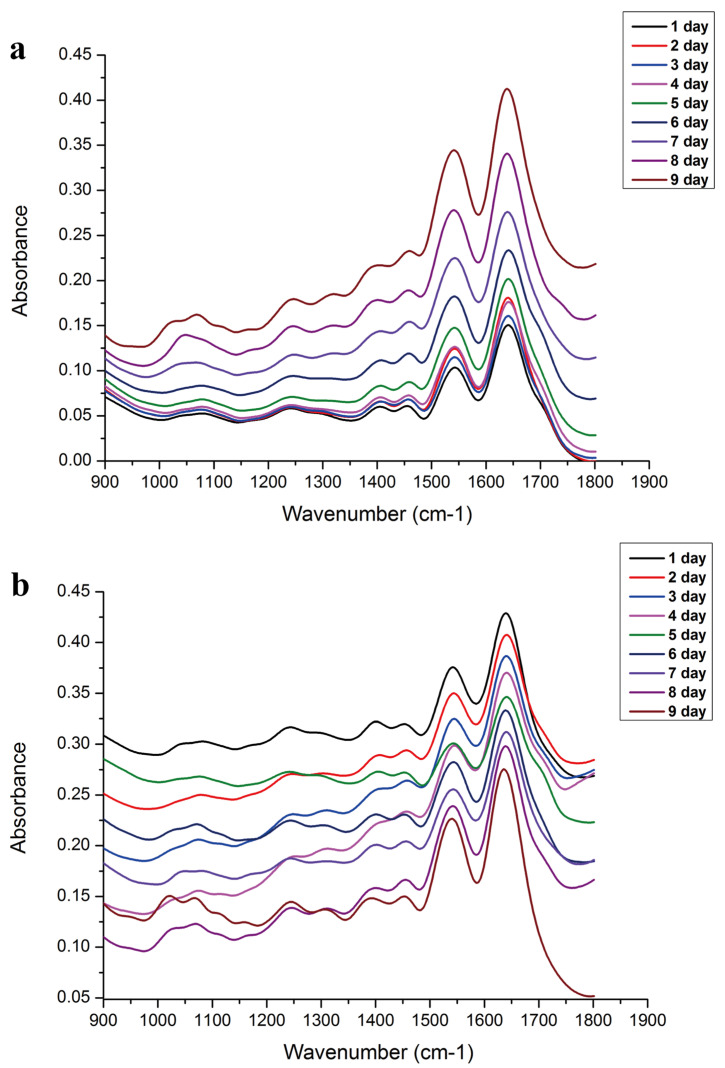
The average FTIR spectra of different intra−puparial development ages (days) in the range of 1800–900 cm^−1^. (**a**) Group A, fluctuating temperature. (**b**) Group C, constant temperature.

**Figure 6 animals-13-01607-f006:**
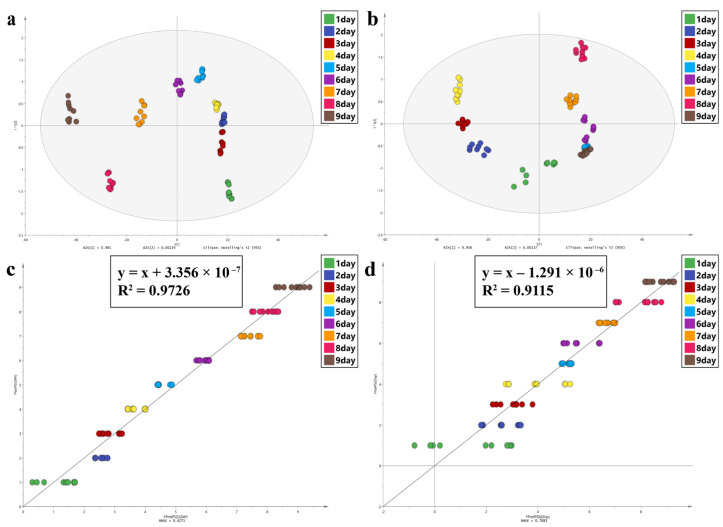
Chemometrics was used to analyze the data that the ATR−FTIR study generated. (**a**,**b**) OPLS-DA analysis of group A and group C, respectively. (**c**,**d**) PLS analysis of group A and group C, respectively.

**Figure 7 animals-13-01607-f007:**
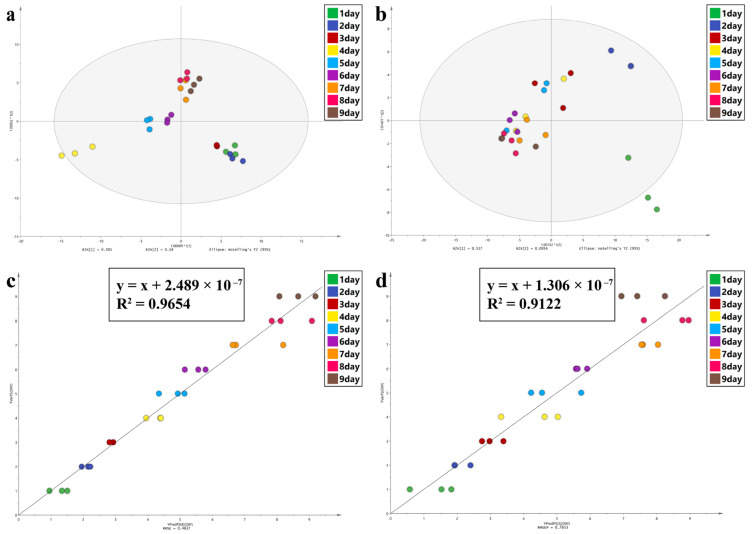
Chemometrics was used to analyze the data that the CHCs study generated. (**a**,**b**) OPLS−DA analysis of group A and group C, respectively. (**c**,**d**) PLS analysis of group A and group C, respectively.

**Table 1 animals-13-01607-t001:** The relationship between the body length (y) (mm) and the time after larviposition (x) (h) of *S. peregrina* larvae at constant and fluctuating temperatures was described using equations. *p* values, degrees of freedom (df), and coefficients of determination (R^2^) are listed.

Temperature (°C)	Equation	df	*p*	R^2^
Fluctuating (A)	y = 4.21804 − 0.24142 × x+ 0.01422 × x^2^ − 1.55605 × 10^-4^ × x^3^ + 4.88045 × 10^-7^ × x^4^	15	<0.001	0.99118
Fluctuating (B)	y = 3.92763 − 0.21602 × x + 0.01444 × x^2^ − 1.67742 × 10^-4^ × x^3^ + 5.61143 × 10^-7^ × x^4^	13	<0.001	0.99808
Constant(C)	y = 3.70744 − 0.20709 × x + 0.0174 × x^2^ − 2.25212 × 10^-4^ × x^3^ + 8.25105 × 10^-7^ × x^4^	11	<0.001	0.99991

**Table 2 animals-13-01607-t002:** Mean (±SD) development duration (h) of *S. peregrina* at constant and fluctuating temperatures.

Developmental Stages	FirstInstar	SecondInstar	ThirdInstar	Wandering	Pupariation	Total Duration
Fluctuating (A)	23.7 ± 2.51	27.7 ± 3.51	70.6 ± 5.1	37.6 ± 5.85	223 ± 11.35	382.6 ± 9.07
Fluctuating (B)	22.6 ± 2.57	25.6 ± 2.08	67.6 ± 7.2	35.6 ± 4.5	218.6 ± 5.5	370.3 ± 10.7
Constant (C)	22.3 ± 2.08	23.3 ± 2.08	65.3 ± 4.1	30.3 ± 2.08	217.3 ± 6.1	358.6 ± 15.04

**Table 3 animals-13-01607-t003:** The relationship between the relative gene expression (y) (fold change) and the intra-puparial age (x) (d) of *S. peregrine* at different temperatures (group A and group C) are described using simulation equations. F values, *p* values, and the coefficient of determination (R^2^) are listed.

Gene	Temperature (°C)	Simulation Equation	F	*p*	R^2^
circRNA_2143	A	y = 1.3774 + 1.04846 × x+ 0.81578 × x^2^ − 0.30251 × x^3^ + 0.02911 × x^4^	4.52766	<0.001	0.93603
	C	y = 7.63272 − 12.85128 × x + 6.8054 × x^2^ − 1.21003 × x^3^ + 0.07401 × x^4^	5.33739	<0.001	0.92318
circRNA_3489	A	y = 0.46214 + 2.31708 × x − 1.10734 × x^2^ + 0.17956 × x^3^ − 0.00686 × x^4^	0.37142	<0.001	0.98587
	C	y = 2.22275 − 3.19588 × x + 1.62495 × x^2^ − 0.31026 × x^3^ + 0.02049 × x^4^	4.67951	<0.001	0.9675
−circRNA_2847	A	y = − 4.19518 + 8.98938 × x − 3.86827 × x^2^ + 0.60917 × x^3^ − 0.03011 × x^4^	0.41914	<0.001	0.98452
	C	y = − 3.59782 + 5.57439 × x − 2.12695 × x^2^ + 0.31238 × x^3^ − 0.01444 × x^4^	1.89332	<0.001	0.97677
fln	A	y = 12.56685 − 29.36167 × x + 23.64982 × x^2^ − 7.276 × x^3^ + 0.77481 × x^4^	3.52198	<0.001	0.96925
	C	y = − 20.85524 + 38.63413 × x − 21.49614 × x^2^ + 3.80815 × x^3^ − 0.05313 × x^4^	8.86323	<0.001	0.91811
UQCRFS1	A	y = 2.57914 − 2.57384 × x + 1.15168 × x^2^ − 0.2 × x^3^ + 0.01229 × x^4^	1.78359	<0.001	0.94842
	C	y = 6.03845 − 4.55755 × x + 1.42931 × x^2^ − 0.20105 × x^3^ + 0.0112 × x^4^	1.34105	<0.001	0.98228
COX5A	A	y = 2.39587 − 2.59247 × x + 1.44986 × x^2^ − 0.29885 × x^3^ + 0.02108 × x^4^	2.5781	<0.001	0.93142
	C	y = 10.4182 − 11.11295 × x + 4.40356 × x^2^ − 0.72446 × x^3^ + 0.04283 × x^4^	4.32924	<0.001	0.89024

**Table 4 animals-13-01607-t004:** FTIR spectroscopy identification of major characteristic peaks.

Baseline Points (cm^−1^)	Wavenumber (cm^−1^)	Infrared Band
1760~1730	1740	Lipid (C = O stretching vibration)
1680~1610	1640	Amide I (C = O stretching)
1580~1510	1544	Amide II (N-H bending coupled to C-N stretching)
1480~1420	1458	C–H bending from CH_2_ and CH_3_
1420~1350	1405	C=O vibrations of COO− from free fatty acids, free amino acids and polypeptides
1330–1277	1309	Amide III
1245–1230	1241	CH_3_–CO Symmetric stretching
1161–1095	1121	C–O, C–OH and P–O vibration
1083–1078	1080	PO_2_-symmetric stretching
1100~1000	1041	C–O(H) stretching vibration
945~906	927	C–O or C–OH vibrations from carbohydrates

**Table 5 animals-13-01607-t005:** Model evaluation parameters and regression equations.

**Model Evaluation Parameters and Regression Equations of ATR-FTIR Study**						
**PLS**				**OPLS-DA**		
	Equation	R^2^	RMSE(DAY)	R^2^X (cum)	R^2^Y (cum)	Q^2^ (cum)
A	y = x + 3.356 × 10^−7^	0.9726	0.4271	1	0.801	0.793
C	y = 1x − 1.2916 × 10^−6^	0.9115	0.7683	1	0.744	0.707
**Model evaluation parameters and regression equations of CHCs study**						
**PLS**				**OPLS-DA**		
	Equation	R^2^	RMSE(DAY)	R^2^X (cum)	R^2^Y (cum)	Q^2^ (cum)
A	y = x + 2.489e × 10^−7^	0.9654	0.4837	0.973	0.927	0.551
C	y = x + 1.306 × 10^−7^	0.9122	0.7653	0.949	0.824	0.44

**Table 6 animals-13-01607-t006:** The distribution of three compound classes of *S. peregrina* intra-puparial tissue at constant and fluctuating temperatures.

	**Fluctuating Temperature (Group A)**								
**Compounds**	**1 Day**	**2 Day**	**3 Day**	**4 Day**	**5 Day**	**6 Day**	**7 Day**	**8 Day**	**9 Day**
n-alkanes(**21**) *	40.92%	28.11%	31.42%	39.46%	39.66%	41.51%	43.87%	45.69%	51.19%
branched alkanes(17) *	55.61%	67.85%	63.46%	55.47%	56.88%	54.83%	50.41%	46.16%	39.54%
alkenes(3) *	3.47%	4.04%	5.12%	5.07%	3.46%	3.66%	5.72%	8.14%	9.27%
	**Constant temperature (group C)**								
**Compounds**	**1 day**	**2 day**	**3 day**	**4 day**	**5 day**	**6 day**	**7 day**	**8 day**	**9 day**
n-alkanes(21) *	41.54%	31.99%	31.05%	38.17%	27.79%	37.42%	38.40%	39.28%	47.20%
branched alkanes(17) *	52.93%	63.94%	64.87%	58.40%	68.29%	58.51%	53.48%	49.31%	40.68%
alkenes(3) *	5.52%	4.07%	4.08%	3.43%	3.92%	4.07%	8.12%	11.41%	12.11%

* The number of each compound.

## Data Availability

All raw data involved in this study are kept in Guo’s laboratory. To request the data, please contact the corresponding author of the article.

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
