# Peer review of "Development of Forensically Important Sarcophaga peregrina (Diptera: Sarcophagidae) and Intra-Puparial Age Estimation Utilizing Multiple Methods at Constant and Fluctuating Temperatures"

_animals, 2023, doi:10.3390/ani13101607_

Round 1

Reviewer 1 Report

The manuscript deals with an interesting aspect related to forensic entomology applied to PMI estimation. In this regard, results are interesting but refer just to certain temperature conditions. Then the age of the pupae, in terms of days or hours, cannot be estimated but under the same conditions, which diminish the general utility. Nevertheless, results are interesting since show how some biological indicators can be useful for such purposes when wider knowledge of the biology of the species be available.

Author Response

Dear Reviewer,

Thank you for taking the time to review our manuscript entitled “Development of Forensically Important Sarcophaga peregrina (Diptera: Sarcophagidae) and Intra-Puparial Age Estimation Utilizing Multiple Methods at Constant and Fluctuating Temperatures” (Manuscript ID: animals-2310679) and for your valuable feedback. We appreciate your time and expertise in reviewing our research. We are committed to conducting further research in this field and look forward to contributing more in the future.

Thank you again for your support.

Best regards,

Yours sincerely,

Yadong Guo

Institution: School of Basic Medical sciences, Central South University, Changsha 410013, Hunan, China

Reviewer 2 Report

Excellent and valuable paper which considers a variety of measures to assess age in a forensically important insect. Well researched and well analyzed. 

The authors examine the effects of constant and fluctuating temperatures on a forensically important species and also one for whom there is not a great deal of data. They compare constant and fluctuating temperatures which is important as most studies look at constant temperatures, yet in the real world, temperatures are not constant. They also look specifically at intra puparial development which is very rarely done and much more difficult. They also use a variety of methods to assess the impacts

The topic is original or relevant in the field, because, as mentioned above, this is a forensically important species, yet very few studies look at sarcophagids, most look at calliphorids. Calliphorids are more common in many areas on cadavers, but some geographic regions have many sarcophagids for whom there is little data. As well, it looks beyond the usual observations of development and also considered other methods of assessment such as gene expression, cuticular hydrocarbons and spectroscopy which greatly adds to its value and suggests new methods of assessment

This manuscript provides a comparison of the effects of constant and fluctuating temperatures, importantly showing that, in real world situations, this species takes longer to develop. This is very important in a forensic context. They also assess the intra-puparial time which is hard to assess and yet vital as it covers almost half of the immature developmental time. Their use of more novel assessment tools is intriguing and very useful moving forward.

The results and the conclusions are clear and appear solid, and the conclusions are consistent with the evidence and arguments presented

The references are appropriate.

Author Response

Dear Reviewer,

Thank you for taking the time to review our manuscript entitled “Development of Forensically Important Sarcophaga peregrina (Diptera: Sarcophagidae) and Intra-Puparial Age Estimation Utilizing Multiple Methods at Constant and Fluctuating Temperatures” (Manuscript ID: animals-2310679) and for your kind words of recognition. We are thrilled to hear that our research has made a positive impact in the field and we greatly appreciate your thoughtful feedback.

We are committed to furthering our understanding of this topic and will continue to conduct research in this area. Your feedback has been invaluable in guiding our future work and we will take your suggestions into consideration as we move forward.

We are committed to learning more about this topic and will continue our research in this area. We hope to promote the use of forensic entomological evidence in the judicial system. Your feedback greatly encouraged us. Once again, thank you for taking the time to review our work.

Best regards,

Yours sincerely,

Yadong Guo

Institution: School of Basic Medical sciences, Central South University, Changsha 410013, Hunan, China

Reviewer 3 Report

Minor Comments

L. 14 – Remove the first word “The” from this sentence.

L. 15 and 16 – Remove “using growth and development data of necrophagous insects”.

L. 15 – Change “postmortem interval” to “minimum postmortem interval” to remains consistent with terminology used in the rest of the manuscript.

L. 16 and L. 20 – Italicize S. peregrina.

L. 41 – Change “were” to “are”.

L. 42 – Change “was estimated” to “is estimated”.

L. 40 – 44 – Need appropriate references to back up statements.

L. 45 – 46 – Need appropriate references to back up statements.

L. 47 – Remove “the” between “on” and “constant temperatures”.

L. 47 – Rephrase sentence to something like: “These studies focused on several necrophagous Dipterans (e.g., C. megacephala, S. dux, etc….), Coleoptera (e.g., N. rufipes), and Hymenoptera (e.g., N. vitripennis).

L. 52 – 68 – Combine this paragraph with the paragraph immediately before.

L. 79 – Change “blowflies” to “blow flies”.

L. 94 – Change “researches” to “research”.

L. 95 – “A few studies have concentrated on the effects of fluctuating temperatures on S. peregrina development”. Please cite these studies.

L. 101 – PMI estimation or PMImin estimation? There is a difference.

L. 120 – “Pig lung was used to induce larvae to hatch”. Does S. peregrina oviposit or larviposit? If they oviposit, then the sentence should be rephrased to “…induce eggs to hatch”.

L. 121 – 122 – Were larvae randomly sampled?

Author Response

Dear Reviewer,

Thank you for taking the time to review our manuscript entitled “Development of Forensically Important Sarcophaga peregrina (Diptera: Sarcophagidae) and Intra-Puparial Age Estimation Utilizing Multiple Methods at Constant and Fluctuating Temperatures” (Manuscript ID: animals-2310679) and for your valuable feedback.

We appreciate your insights and suggestions, which have helped us improve the quality of our work. We have completed the revisions to the manuscript in response to your comments. In response to some of your comments, we would like to explain:

Comment 1:

  1. 95 – “A few studies have concentrated on the effects of fluctuating temperatures on S. peregrina development”. Please cite these studies.

Answer:

Our literature search has not revealed any studies on the effect of fluctuating temperatures on the development of S. peregrina. Since we cannot exclude that our search was incomplete, we wanted to state "almost no relevant studies" at first, but for the sake of rigor and consideration for your comments, we now state, “Until this study, we have not searched the literature on the effect of fluctuating temperature on the development of S. peregrina through commonly used literature search platforms (PubMed and Web of Science)”.

Comment 2:

  1. 120 – “Pig lung was used to induce larvae to hatch”. Does S. peregrina oviposit or larviposit? If they oviposit, then the sentence should be rephrased to “…induce eggs to hatch”.

Answer:

To the best of our knowledge, S. peregrina exhibits ovoviviparity. During our observation of the rearing process under most conditions, it produced larvae. However, it will lay eggs when conditions are unfavorable, such as when the substrate is too dry, when temperature and humidity are abnormal, or during unsuitable breeding seasons such as winter. In this experiment, conditions were suitable, and larvae were produced. Hence, we state, “Pig lung was used to induce larvae to hatch”.

Comment 3:

  1. 121 – 122 – Were larvae randomly sampled?

Answer:

Yes, they are randomly sampled. We have added this point to the manuscript.

We hope that the changes we have made will meet your requests. We believe that the changes we have made have strengthened the manuscript. Once again, we appreciate your feedback.

Best regards,

Yours sincerely,

Yadong Guo

Institution: School of Basic Medical sciences, Central South University, Changsha 410013, Hunan, China

Reviewer 4 Report

There were a few minor  issues, but the paper itself is sound. It's also an important and interesting concept and needs  to be in the literature. 

Author Response

Dear Reviewer,

Thank you for taking the time to read our manuscript entitled “Development of Forensically Important Sarcophaga peregrina (Diptera: Sarcophagidae) and Intra-Puparial Age Estimation Utilizing Multiple Methods at Constant and Fluctuating Temperatures” (Manuscript ID: animals-2310679) and for providing us with your valuable feedback. We appreciate your insights and suggestions, which have helped us to improve the quality of our work.

We have completed all revisions to the manuscript and have added relevant references. We believe that the changes we have made have strengthened the manuscript and addressed the issues you raised.

We also wanted to express our gratitude for your unique review style, particularly your comment, "Oh, I'm excited about this paper!" This is refreshing, and we appreciate your recognizing our work.

Once again, thank you for your time and expertise in reviewing our manuscript. Our current and future research aims to enhance the utilization of forensic entomological evidence in cases involving death. We look forward to future opportunities to contribute to the field.

Best regards,

Yours sincerely,

Yadong Guo

Institution: School of Basic Medical sciences, Central South University, Changsha 410013, Hunan, China
